# Use of Measurable Residual Disease to Evolve Transplant Policy in Acute Myeloid Leukemia: A 20-Year Monocentric Observation

**DOI:** 10.3390/cancers13051083

**Published:** 2021-03-03

**Authors:** Francesco Buccisano, Raffaele Palmieri, Alfonso Piciocchi, Luca Maurillo, Maria Ilaria Del Principe, Giovangiacinto Paterno, Stefano Soddu, Raffaella Cerretti, Gottardo De Angelis, Benedetta Mariotti, Maria Antonietta Irno Consalvo, Consuelo Conti, Daniela Fraboni, Mariadomenica Divona, Tiziana Ottone, Serena Lavorgna, Paola Panetta, Maria Teresa Voso, William Arcese, Adriano Venditti

**Affiliations:** 1Department of Biomedicine and Prevention, University Tor Vergata of Roma, 00133 Rome, Italy; raffaele.palmieri@ptvonline.it (R.P.); luca.maurillo@uniroma2.it (L.M.); del.principe@med.uniroma2.it (M.I.D.P.); giovangiacinto.paterno@ptvonline.it (G.P.); raffaella.cerretti@libero.it (R.C.); gottardo.deangelis@tiscali.it (G.D.A.); benedetta.mariotti@ptvonline.it (B.M.); mirnoconsalvo@scamilloforlanini.rm.it (M.A.I.C.); consuelo.conti@ptvonline.it (C.C.); danielafraboni@virgilio.it (D.F.); mariadomenica.divona@ptvonline.it (M.D.); tiziana.ottone@uniroma2.it (T.O.); serena.lavorgna@uniroma2.it (S.L.); paola.panetta@ptvonline.it (P.P.); voso@uniroma2.it (M.T.V.); william.arcese@uniroma2.it (W.A.); adriano.venditti@uniroma2.it (A.V.); 2Centro Dati Fondazione GIMEMA, 00100 Rome, Italy; a.piciocchi@gimema.it (A.P.); s.soddu@gimema.it (S.S.); 3Rome Transplant Network, Tor Vergata University Hospital, 00133 Rome, Italy

**Keywords:** biomarkers, AML, MRD, cytogenetics and molecular markers, multiparametric flow cytometry

## Abstract

**Simple Summary:**

Upfront genetics/cytogenetics and minimal measurable disease (MRD) are becoming relevant biomarkers in the process of post-remission transplant allocation in AML. However, until recently the transplantation choice relied on the availability of a fully matched familiar donor, whereas individual patient- and disease-related characteristics played a secondary role in transplant allocation. In this paper we analyzed the evolution of the transplantation policy at our center in a 20-year time interval. At the beginning of our observation patients were submitted to allogeneic transplant, per protocol, mostly if a fully matched family donor was available or to autologous transplant if no fully matched family donor was identified (“donor vs. no donor” strategy) regardless of upfront genetics/cytogenetics or MRD status. Thereafter, persistence of MRD after consolidation cycle was included in the decision-making process for transplant selection. Patients with favorable and intermediate-risk cytogenetic risk were to receive allogeneic or autologous stem cell transplantation if MRD positive or negative, respectively, (“transplant vs. no transplant” strategy). In this cohort, patients with FLT3-ITD or adverse risk karyotype were submitted to allogeneic transplant as well.

**Abstract:**

Measurable residual disease (MRD) is increasingly employed as a biomarker of quality of complete remission (CR) in intensively treated acute myeloid leukemia (AML) patients. We evaluated if a MRD-driven transplant policy improved outcome as compared to a policy solely relying on a familiar donor availability. High-risk patients (adverse karyotype, FLT3-ITD) received allogeneic hematopoietic cell transplant (alloHCT) whereas for intermediate and low risk ones (CBF-AML and NPM1-mutated), alloHCT or autologous SCT was delivered depending on the post-consolidation measurable residual disease (MRD) status, as assessed by flow cytometry. For comparison, we analyzed a matched historical cohort of patients in whom alloHCT was delivered based on the sole availability of a matched sibling donor. Ten-years overall and disease-free survival were longer in the MRD-driven cohort as compared to the historical cohort (47.7% vs. 28.7%, *p* = 0.012 and 42.0% vs. 19.5%, *p* = 0.0003). The favorable impact of this MRD-driven strategy was evident for the intermediate-risk category, particularly for MRD positive patients. In the low-risk category, the significantly lower CIR of the MRD-driven cohort did not translate into a survival advantage. In conclusion, a MRD-driven transplant allocation may play a better role than the one based on the simple donor availability. This approach determines a superior outcome of intermediate-risk patients whereat in low-risk ones a careful evaluation is needed for transplant allocation.

## 1. Introduction

Nowadays, allogeneic hematopoietic cell transplantation (alloHCT) represents the most effective antileukemic treatment and it is increasingly employed in patients affected with acute myeloid leukemia (AML) up to the age of 70 yrs [1]. In spite of this, overall outcome of adult patients remains unsatisfactory with only 40% of young patients and less than 20% of older patients being cured of their disease [2]. 

Meta-analyses of large prospective studies indicate that, whatever the underlying genetic/cytogenetic abnormality, the beneficial effect of alloHCT takes place as soon as the risk of relapse exceeds 35–40%; when probabilities of relapse are below those percentages the risk of non-relapse mortality (NRM) will attenuate the survival advantage of alloHCT [3,4,5]. Therefore, delivery of alloHCT in first complete remission (CR1) appears inappropriate for patients who can experience a long-term survival such as those with CBF-AML, biallelic CEBPA-mutated and NPM1-mutated AML. On the other hand, in high-risk patients, e.g., those with complex or monosomic karyotype or poor-risk gene mutations, alloHCT in CR1 appears mandatory [6].

Nevertheless, the increasing knowledge of the genetic landscape of AML has shed light on the complex gene interplay affecting prognosis in AML and some disease categories (e.g., RUNX1-RUNX1T1 and NPM1 mutated AML) have resulted to be more complex than expected with a final outcome that may eventually be not as favorable as expected [7]. If these categories might take advantage of early intensive post-remission treatments, e.g., alloHCT, needs to be carefully evaluated and it is still matter of debate [4]. Furthermore, there are no accepted criteria to direct the decision-making process after consolidation for patients in the intermediate-risk category: for these patients, search for biomarkers allowing to extrapolate those at high or low risk of relapse is warranted [8]. 

In this view, measurable (previously minimal) residual disease (MRD) promises to be a robust biomarker in that it captures the diversities of the different genetic/cytogenetic features of AML and recapitulates other patient-specific heterogeneities regarding drug bioavailability, metabolism and chemoresistance [9,10]. Once a chemotherapy-induced CR is achieved, MRD assessment at several time-points has shown to predict outcome, potentially paving the way to treatment adjustments. For the first time, the recent updated recommendations of European LeukemiaNet (ELN) adopted the concept of MRD negative CR. This is the CR in which a given genetic or immunophenotypic aberrancy, present at diagnosis, is no longer detectable by high sensitivity multiparametric flow-cytometry (MFC) or reverse transcriptase-quantitative PCR (RT-qPCR) [11]. MRD may be especially informative in patients lacking specific molecular risk profile (e.g., intermediate risk karyotype) or in those in whom the prognostic impact of specific gene mutations may be modified by other concurrent mutations (e.g., FLT3 mutations) [7]. 

We evaluated if MRD positivity after consolidation might allow to identify patients with high-risk of disease recurrence among those without favorable or intermediate genetics/cytogenetics (“MRD-driven” strategy) and if an MRD-driven alloHCT in first CR might lead to overall (OS) and disease-free survival (DFS) improvement in these categories of patients. 

Establishing a proper transplant policy was a crucial part of the MRD-driven approach. In fact, once declared at high-risk, patients were to receive alloHCT whatever the source of stem cells: matched sibling donor (MSD), matched unrelated donor (MUD), umbilical cord blood (UCB) or haploidentical related donor (HRD). Well aware of the potential biases deriving from an historical comparison between populations of patients spanning over a 20 years period, we analyzed the outcome of a matched cohort of patients who had in common the same treatment design except being submitted to alloHCT only if a fully matched family donor was available or to autologous stem cell transplant (AuSCT) if not (“donor-driven” strategy), regardless of the baseline risk profile or of the MRD status. 

## 2. Materials and Methods

We analyzed a population of 185 AML consecutive patients affected with de novo non-M3 AML in CR1, diagnosed in the period between 1998 and July 2013 at Tor Vergata University and St. Eugenio Hospital of Rome. In our retrospective analysis, post-consolidation timepoint was identified as the most informative one for MRD prognostication in European Organization for Research and Treatment of Cancer (EORTC)/Gruppo Italiano Malattie Ematologiche Maligne dell’Adulto (GIMEMA) protocols [12]. Patients not in CR at this timepoint or lacking a leukemia-associated immunophenotype (LAIP) suitable for MRD monitoring were not included. This series represents an extension of a cohort of patients already analyzed for different purposes and reported previously [12,13,14,15].

From January 1998 to January 2008, patients were enrolled into the EORTC/GIMEMA AML-10 and AML-12 trials [16,17]. Patients were enrolled provided they met the following criteria for eligibility: (1) age 18 to 60.9 years; (2) AML other than M3; (3) WHO performance status 0–3; (4) adequate liver (serum bilirubin level ≤ 2 UNL; AST and ALT ≤ 3 UNL) and renal (serum creatinine ≤ 2 UNL) functions; (5) LVEF ≥ 50% by echocardiogram; (6) absence of severe concomitant neurological or psychiatric diseases and congestive heart failure or active uncontrolled infections; (7) signed informed consent. Patients with therapy-related AML were not considered eligible. Exclusion criteria included blast crisis of chronic myeloid leukemia, AML supervening after other chronic myeloproliferative diseases or antecedent myelodysplastic syndromes of more than six months duration and other progressive malignant diseases. In the AML10 study [16], remission induction treatment consisted of cytarabine 25 mg/m^2^ as intravenous bolus followed by SDAC 100 mg/m^2^ given as a daily continuous infusion for 10 days; i.v. etoposide 100 mg/m^2^/d on days 1 to 5; and on days 1, 3, and 5 one of the following: daunorubicin 50 mg/m^2^ (DCE arm), mitoxantrone 12 mg/m^2^ (MICE arm), or idarubicin 10 mg/m^2^ (ICE arm). Patients in CR, received a single course of consolidation therapy, consisting in intermediate dose cytarabine (500 mg/m^2^) days 1 to 6, and the same intercalator used in induction, on days 4 to 6. After consolidation, those patients with an HLA-compatible sibling received an alloHCT while those without a sibling donor were randomly assigned to AuSCT. In the AML-12 trial, [17] patients were randomly assigned to high dose cytosine arabinoside (HDARAc, 3 g/m^2^/12 h for four days) plus daunorubicin (50 mg/m^2^/d on days 1,3,5) and etoposide (100 mg/m^2^/d on day 1 to 5) or to DCE arm of AML-10 study. Patients obtaining CR received as the same consolidation regimen as AML-10 trial followed by alloHCT or AuSCT, depending on donor availability. 

Since February 2008 to July 2013, due to conclusion of AML12 trial, patients were treated according to the standard-dose cytarabine AML12 arm. As an induction therapy, the patients received cytosine arabinoside 100 mg/m^2^ as a daily continuous infusion for 10 days; i.v. etoposide 100 mg/m^2^/d on days 1 to 5 and, daunorubicin 50 mg/m^2^ on days 1, 3, and 5. Those in CR after 1 or 2 cycle of induction were given a consolidation therapy with cytosine arabinoside 500 mg/m^2^ twice a day, on days 1 to 6 and daunorubicin 50 mg/m^2^ on days 4 to 6. However, at variance with AML12 protocol, assignment to AuSCT or alloHCT was based on the integrated evaluation of baseline genetics/cytogenetics and immunophenotypic MRD status after consolidation cycle rather than on the donor availability. 

More in detail, patients belonging to the intermediate genetics/cytogenetics were to receive AuSCT and alloHCT if MRD negative or positive, respectively. For those patients FLT3-ITD or bearing adverse karyotype an alloHCT had to be delivered. Patients allocated to AuSCT were to receive HDARAc in the case of stem cell mobilization or collection failure. On the other hand, patients qualified as high-risk were to receive an alloHCT whatever the source of stem cells (MSD, MUD, UCB or HRD). AuSCT or alloHCT were to be delivered within three months from the end of consolidation therapy. Patients enrolled in the GIMEMA AML1310 protocol [18] were not included in the present analysis.

Since the retrospective nature of the study, informed consent for data collection was obtained from all subjects according to the specific per-protocol procedures of the trials involved in the present analysis.

Therefore, for the purposes of the present analysis, we identified two patients’ cohorts: the MRD-driven cohort, in which post-consolidation assignment was risk-stratified as mentioned above, and the donor-driven cohort in which, per AML10/AML12 protocols, transplantation assignment relied only on the availability/lack of a familiar sibling donor, regardless of the individual risk profile (Appendix A). 

### 2.1. Transplantation Regimens

High-dose busulfan combined with cyclophosphamide (BuCy) was the most commonly used conditioning regimen for AuSCT. As for as alloHCT, until the 2008, there were two main conditioning regimens, BuFlu and BuCy. From 2008, we adopted an identical conditioning regimen (Thiotepa-Busilvex-Fludarabine, TBF) for the MSD, MUD or HRD. The conditioning regimen included the anti-thymocyte globulin (ATG-Fresenius^®^) given at dose of 5 mg/kg per day on days − 4 through − 1 in MUD and HRD. Graft-versus-host disease (GVHD) prophylaxis changed according to different sources of hematopoietic stem cells and included cyclosporine (CyA) plus i.v. methotrexate (15 mg/m2 on day +1 and at 10 mg/m^2^ on day 3, 6 and 11) in HLA-matched sibling and MUD. In HRD all patients received the same GVHD prophylaxis. It consisted of a combination of 5 drugs: anti-thymocyte globulin, CyA (from day 180 and in the absence of chronic GVHD, CyA was progressively tapered and stopped by day 365), methotrexate, mycophenolate mofetil (p.o. at 15 mg/kg/d from day 7 to 100). Furthermore, Basiliximab, an anti-CD25 monoclonal antibody, was given at fixed dose of 20 mg on days 0 and 4.

### 2.2. Immunophenotypic Studies and MRD Detection

At diagnosis, bone marrow (BM) samples were collected to perform immunophenotypic, chromosomal, and genetic studies as detailed elsewhere [12,13,14,15]. Cases expressing a LAIP were selected and re-analysed by staining with the relevant combinations of antibodies in a polychromatic assay. This step served to define a leukaemia immunophenotypic fingerprint that, in turn, was used to track residual leukaemia cells (RLCs) in BM collected after each therapeutic step and during follow-up [12,13,14,15]. The polychromatic assay approach also allowed for the identification of ≥2 antibody combinations for each case, therefore minimizing pitfalls due to phenotypic changes possibly occurring upon relapse [19]. As previously reported, the threshold for MRD negativity was set below a number of 3.5 × 10^−4^ RLCs. Genetic/cytogenetic abnormalities were classified according to MRC criteria or NCCN 2009 criteria [20,21].

### 2.3. Statistical Analysis and Outcome Definitions

Differences in the baseline characteristics were compared by χ^2^ test or Fisher’s exact test for categorical data. Relationship of MRD to treatment response was estimated by two-sided χ^2^ or Fisher’s exact test. OS was defined as the time interval from study entry until death from any cause or last follow up for patients alive. RFS was measured from the date of achievement of a remission until the date of relapse or death from any cause; patients not known to have relapsed or died at last follow-up were censored on the date they were last examined [22]. RFS and OS curves were estimated using the Kaplan-Meier product-limit method and compared using the log-rank test. The potential significance of prognostic factors was explored with Cox proportional hazards (PH) models in multivariable analysis. The estimate of the cumulative incidence of relapse (CIR) was obtained by using competing risk methods [23]. All tests were two-sided, accepting *p* < 0.05 as indicating a statistically significant difference. All covariates were evaluated in univariate models and all factors with univariate association within *p*-value < 0.1 were considered in the multivariate models as potential parameters. Backward and stepwise methods were applied to identify the multivariate models with a step-by-step iterative construction that involves the selection of independent variables to be considered in the final model. Analyses were performed by SAS Version 9.2 statistical software (SAS Institute, Cary, NC, USA) and R (R Foundation for Statistical Computing, Vienna, Austria. ISBN 3-900051-07-0).

## 3. Results

Among 185 evaluable patients, 115 (62.2%) belonged to the MRD-driven cohort whereas 70 (37.8%) represented the donor-driven cohort. The clinical characteristics of the patients are detailed in Table 1. The two groups were balanced in terms of sex distribution, White Blood Cell (WBC) count >50 × 10^9/L, frequency of FLT3 mutations and karyotype. Age was slightly higher in the Donor-driven cohort (median 49.5 yrs vs. 44 yrs, *p* = 0.046). We also observed a higher frequency of NPM1 mutated patients in the MRD-driven cohort (74.5% vs. 55.4%, *p* = 0.017). As expected, more patients were addressed to alloHCT in the MRD-driven cohort than in the Donor-driven one (43.0% vs. 5.7%, *p* < 0.001). This was mainly due to the larger use of alternative sources of stem cells. In fact, sources of alloHCT were MSD in 27/49 (55%), MUD in 14/49 (29%) and HRD in 8/49 (16%) in the MRD-driven cohort whereas all four alloHCT in the Donor-driven cohort were MSD.

MRD assessment was the additional biomarker considered for the risk stratification of the MRD-driven cohort. We further analyzed clinical and genetic prognosticators in this cohort according to MRD status (Appendix A). MRD positive cases were, as expected, enriched in FLT3-ITD patients (35.9% vs. 8.1%, *p* = 0.004) and NPM1-mutated (36.1% vs. 10.3%, *p* = 0.008). Accordingly, poor-risk karyotypes were more frequent in MRD positive (0% vs. 12.3%) and good-risk ones in MRD negative (40.5% vs. 9.2%) patients, respectively (overall *p* < 0.001).

In the whole series, 10-yrs OS (47.7% vs. 28.7%, *p* = 0.012) and DFS (42.0% vs. 19.5%, *p* = 0.0003) were longer in the MRD-driven cohort as compared to the donor-driven one (Figure 1A,B). Accordingly, Cumulative Incidence of Relapse (CIR) was significantly lower in the MRD-driven cohort than in the Donor-driven one (40.7% vs. 74.8%, *p* < 0.001) (Figure 1C).

When a subgroup analysis was performed, we observed a different impact of treatment strategy according to the karyotypic risk classes. Intermediate risk (IR) patients represented the category which benefited the most by the use of MRD-driven strategy (Appendix A). In fact, OS (52.0% vs. 20.5%, *p* = 0.0045), DFS (42.8% vs. 13.2%, *p* = 0.0002) and CIR (42.0% vs. 76.3%, *p* = 0.0002) were significantly better for the MRD-driven cohort than the Donor-driven one. In the analysis of the IR patients FLT3-ITD patients were not considered.

The number of patients in IR subgroup allowed us to test the outcome of the two cohorts separately in 68 MRD positive and 30 MRD negative patients. MRD-driven cohort did better both in terms of OS (55.2% vs 14.6%, *p* = 0.0035) and DFS (44.3% vs 9.4%, *p* = 0.0004), with a lower CIR (*p* = 0.002). NRM was not statistically different between the two cohorts (Figure 2). Notably, we did not observe any difference in OS, DFS, CIR and NRM in the IR MRD negative patients (Appendix A).

A survival advantage, although not statistically significant, was also appreciated for patients belonging to the favorable (FR) category. In fact, OS and DFS of FR patients (Figure 3A,B) in the MRD-driven cohort were 61.1% and 53.4% versus 51.9% and 42.4%, respectively, of those in the Donor-driven cohort (*p* = 0.528 and *p* = 0.393, respectively). The lack of statistically significant difference may well be explained by the higher non-relapse mortality (NRM) observed among the MRD-driven cohort (23.7% vs. 0%, *p* = 0.102), likely due to the higher number of alloHCT delivered (Figure 3C). Nevertheless, the FR patients belonging to the MRD-driven cohort also had a significantly lower CIR (22.8% vs 57.6%, *p* = 0.042) (Figure 3D). With regard to adverse risk (AR) category, low figures and the overall unfavorable outcome did not allow to observe any statistical difference between the MRD-driven and Donor-driven cohorts. Nevertheless, the three long-term survivors belonged to the MRD-driven cohort.

Due to the role of MRD assessment in risk attribution, a separate analysis of the MRD impact on OS, DFS, CIR and NRM of every karyotype risk category was performed (summarized in Appendix A). Again, MRD affected mostly DFS patients belonging to FR and IR whereas CIR was affected only in IR patients. 

All the variables with a statistical significance in univariable analysis both for OS (Table 2) and DFS (Table 3) were challenged into a multivariable model if resulting independent by a step-by-step iterative model to avoid the inclusion of collinear variables. Transplantation was considered as a time-dependent covariate. This analysis confirmed treatment strategy (MRD-driven vs. donor-driven) as a factor independently affecting OS in both univariable and in multivariable analysis, together with karyotype (Table 2). Differently, although MRD-driven therapy was statistically significant for DFS in univariable analysis, non-allogeneic post-treatment strategy (AuSCT or no SCT vs. alloHCT) and karyotype (IR or AR vs. FR) were the only significant variables in multivariate analysis (Table 3).

## 4. Discussion

In our 20-year retrospective analysis, we demonstrated that the evolution to a comprehensive transplant selection policy, based on the integration of upfront genetics/cytogenetics and post-consolidation MRD assessment (MRD-driven therapy) improves outcome of adult patients with AML as compared to a Donor-driven approach.

The critical role of prognostic biomarkers emerges especially when it comes to deciding as to whether a transplant should be delivered or not in first CR. In this view, the role of MRD has been a subject of discussion for a long time [24]. Although the presence of pre-transplant MRD, measured either by MFC or RT-qPCR, was proven to be a poor prognosticator [25,26], several authors claim that the negative prognostic impact of MRD persistence before alloHCT is counteracted by GVL effect, so that MRD positivity before alloHCT should not be a contraindication to deliver the transplant procedure. This has been consistently confirmed in MFC [27,28] and RT-qPCR studies [29,30]. A recent retrospective analysis demonstrated that the GVL effect occurred in an equally effective manner in MRD positive and negative patients [27]. Accordingly, the authors observed a 63% reduction of the relapse risk but the impact of non-relapse mortality led to a survival advantage only in MRD positive patients. Same observations were reported in CBF-AML and NPM mutated AML [29,30]. 

In our series, the superior outcome of IR patients belonging to the MRD-driven cohort is related to the high frequency of alloHCT delivery to patients who were MRD positive after consolidation cycle. IR category represents a clinical conundrum due to the genetic heterogeneity and the lack of definitive recommendations/guidelines about the transplant choice vs. alternative post remission approaches. In our former analysis, we observed that patients with a level of MRD ≥3.5 × 10^−4^ RLC after consolidation therapy had a worse outcome as compared to those MRD negative (<3.5 × 10^−4^ RLC). Submitting MRD positive patients to alloHCT, improved their prognosis in a way that the survival estimates equalized those of MRD negative ones submitted to chemotherapy and/or AuSCT [12,15]. Therefore, the MRD-driven approach appears a very efficient tool to optimize the use of alloHCT, based on the level of MRD (positive vs negative). In the present series, a purposeful allocation to alloHCT of MRD positive patients allowed an actual improvement of the outcome, whereas, not unexpectedly, such advantage was not observed among MRD negative ones (Figure 2 and Appendix A). Following these retrospective results, GIMEMA has finalized a trial (AML1310) intended to explore whether the prospective delivery of a post remission therapy (alloHCT vs AuSCT), intensity of which is risk-driven (genetics/cytogenetics plus post-consolidation MRD), results in an increased anti-leukemic efficacy and reduced therapy-related toxicity. In the AML1310 trial, the negative effect of MRD positivity in IR patients was abrogated thanks to the prospective allocation to alloHCT [18]. Altogether, our retrospective and prospective experience, suggests that the status of MRD positivity before alloHCT does not represent, especially in IR patients, a contraindication to alloHCT, which remains the most effective approach in this poor prognosis setting [28,31,32]. In our opinion, such a conception appears even more solid in a time when the availability of new, non-chemotherapeutic agents will theoretically allow to design specific transplant protocols, including maintenance, to prevent relapse in these pre-alloHCT MRD positive patients [33]. The idea to resume maintenance strategies appears particularly appealing in situations, such as the presence of complex or monosomic karyotype, TP53 mutations or high/very high pre-transplant MRD levels, in which GVL alone may fail to keep residual AML under control [34]. This may well explain why our AR category did not benefit from alloHCT.

Based on our experience, selecting the optimal post-remission regimen for FR patients (CBF-AML or NPM1 positive AML) poses several challenges. In the past, according to the “one size fits all” policy even patients with CBF-AML were submitted to alloHCT if a matched-sibling donor was available. With the time, it became evident that such an approach did not translate in an improved outcome since NRM counteracted the survival advantage determined by the procedure and the GVL [35]. In a more modern vision, patients with a risk of relapse inferior to 35–40% such as those with CBF-AML and NPM1 mutated AML, are considered to have a favorable course so that alloHCT is not recommended in CR1 [3,4,11]. However, recent MRD studies conducted using RT-qPCR made very clear that even in CBF/NPM1 positive AML the persistence of MRD at any time point during therapy significantly reduce the duration of response and survival [29,30]. This means that in FR AML the role of MRD assessment can be at least as influential as in IR since the status of MRD positivity can convert an initial favorable outlook in an adverse one. In our series, the patients with FR AML who received AuSCT in the MRD-driven cohort had a statistically significant higher rate of CIR (Figure 3C) as compared to those submitted to alloHCT. Indeed, all patients receiving AuSCT were MRD positive by MFC. For these patients, alloHCT might have been a more appropriate solution than AuSCT, in the attempt to gain a better disease control. On the other hand, an excess of NRM (Figure 3D) may have nullified the advantage of this better disease control so that no survival advantage was observed.

## 5. Conclusions

A strategy of MRD-driven post-remission therapy determined an improved OS and DFS as compared to a non-risk-adapted strategy. Such an approach implies that alloHCT is delivered based on the actual risk of disease recurrence, according to a “transplant vs no transplant” rather than “donor versus no donor” policy. This, in turn, implies that once a given patient qualifies as high-risk, alloHCT is to be timely delivered whatever the source of stem cells. In FR patients, AuSCT may be a suboptimal choice if a status of MRD negativity is not reached.

## Figures and Tables

**Figure 1 cancers-13-01083-f001:**
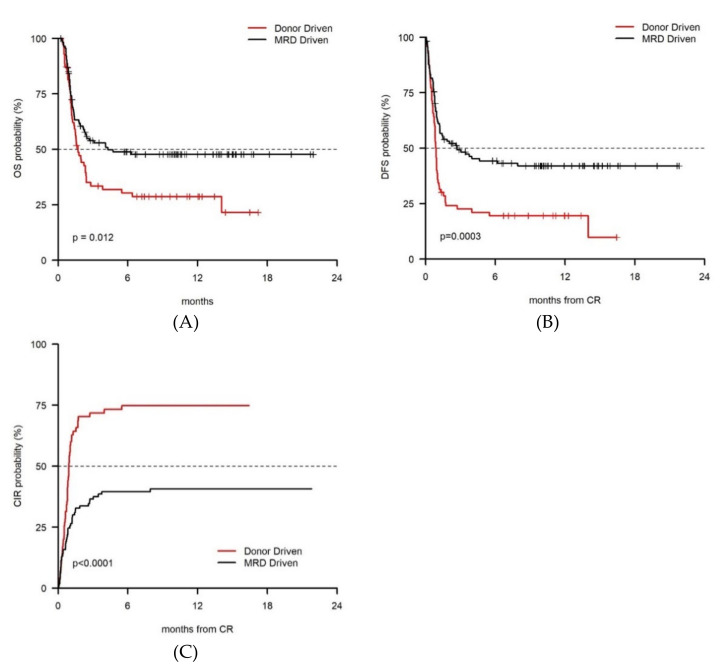
In the whole series, 10-yrs OS (47.7% vs. 28.7%, *p* = 0.012) and DFS (42.0% vs. 19.5%, *p* = 0.0003) were longer for 115 patients in the MRD-driven cohort as compared to 70 patients in the donor-driven cohort (**A**,**B**). Accordingly, CIR (40.7% vs. 74.8%, *p* < 0.001) was lower in the MRD-driven cohort (**C**).

**Figure 2 cancers-13-01083-f002:**
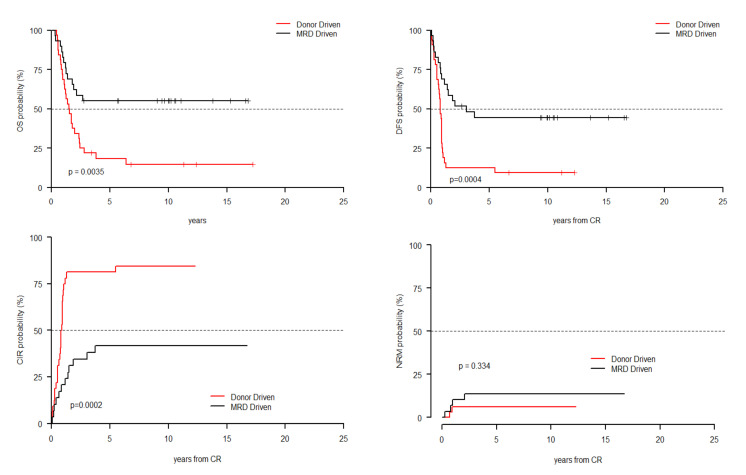
Clinical outcome of 68 MRD positive patients according to cohort allocation. MRD-driven cohort did better both in terms of OS (55.2% vs. 14.6%, *p* = 0.0035) and DFS (44.3% vs. 9.4%, *p* = 0.0004), with a lower CIR (*p* = 0.002). NRM was not statistically different between the two cohorts.

**Figure 3 cancers-13-01083-f003:**
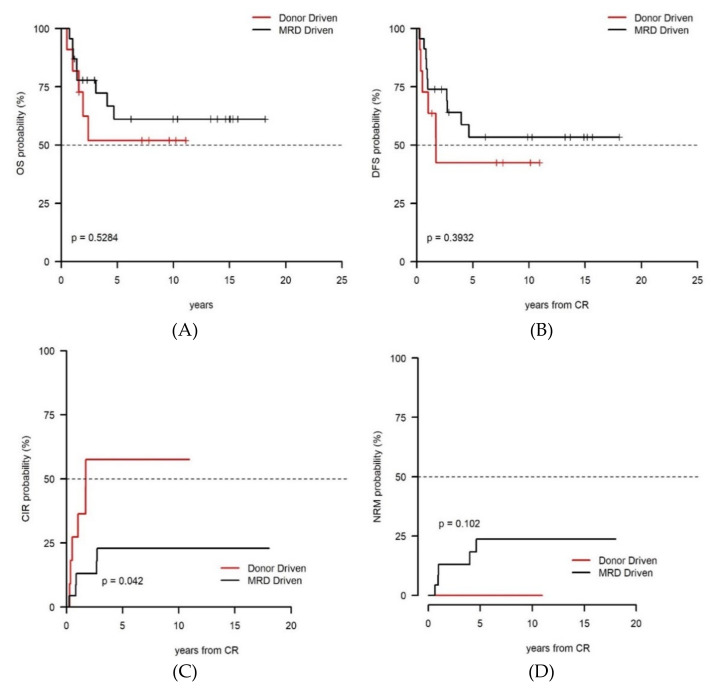
In favorable-risk karyotype MRD-oriented cohort had a better, even if not significant, outcome as compared the donor-oriented cohort. OS (**A**) and DFS (**B**) in the MRD-driven cohort were 61.1% and 53.4% versus 51.9% and 42.4%, respectively, of those in the Donor-driven cohort (*p* = 0.528 and *p* = 0.393, respectively). The superior outcome of the MRD-oriented cohort was mainly due (**C**) to a significantly lower CIR (22.8% vs 57.6%, *p* = 0.042) that however was counterbalanced (**D**) by a higher NRM in the MRD driven cohort due to the higher alloHCT delivery.

**Table 1 cancers-13-01083-t001:** Clinical characteristics of patients belonging to MRD-driven and donor-driven cohorts.

Parameter	Level	Donor-Driven Cohort (*n* = 70, 37.8%)	MRD-Driven Cohort (*n* = 115, 62.2%)	*p*-Value
Sex (%)	Female	29 (41.4)	47 (40.9)	1.000
	Male	41 (58.6)	68 (59.1)	
Age (median [range])		49.50 [17.00, 72.00]	44.00 [18.00, 64.00]	0.046
WBC count (%)	<50.0 × 10^9^/L	45 (64.3)	88 (76.5)	0.104
	>50.0 × 10^9^/L	25 (35.7)	27 (23.5)	
FLT3-ITD (%)	FLT3 negative	53 (80.3)	75 (72.8)	0.356
	FLT3 positive	13 (19.7)	28 (27.2)	
NPM1-mutated (%)	NPM negative	36 (55.4)	76 (74.5)	0.017
	NPM positive	29 (44.6)	26 (25.5)	
Karyotype (%)	Favorable risk *	11 (16.4)	23 (21.1)	0.514
	Intermediate risk *	53 (79.1)	78 (71.6)	
	Adverse risk *	3 (4.5)	8 (7.3)	
Post-remission therapy	alloHCT	4 (5.7)	49 (43.0)	<0.001
(%)	AuSCT	35 (50.0)	37 (32.5)	
	No SCT	31 (44.3)	28 (24.6)	

Abbreviation: WBC = white blood cells; SCT = stem cell transplantation; AuSCT = Autologous stem cell transplantation; alloHCT = Allogeneic stem cell transplantation. * Patients were stratified according to Refined Medical Research Council (MRC) classification of cytogenetic risk, as follows: “favorable” risk [cases with t(8;21), t(15;17) or inv(16)/t(16;16)]; “adverse” risk [cases with complex cytogenetic changes (>3 unrelated abnormalities), −5, add(5q)/del(5q), −7/add(7q), t(6;11), t(10;11), t(9;22), −17, abn(17p) with other changes, 3q abnormalities excluding t(3;5), inv(3)/t(3;3)]; and “intermediate” risk [cases with normal karyotype and other non-complex]; WBCc: white blood cells count; ITD: internal tandem duplication.

**Table 2 cancers-13-01083-t002:** Univariate and multivariate analysis for Overall Survival.

Parameter	Univariate Analysis				Multivariate Analysis			
	HR	Lower 95% CI	Higher 95% CI	*p*	HR	Lower 95% CI	Higher 95% CI	*p*
Treatment strategy (MRD-oriented vs. donor-oriented)	0.62	0.42	0.9	0.0129	0.587	0.396	0.872	0.008
Sex (male vs. female)	1.08	0.73	1.59	0.7129				
Age (>60 vs. <60 years)	1.02	1	1.03	0.067				
WBC count (× 10^9^/L) (>50.0 vs. <50.0)	1.589	1.0618	2.3771	0.0243				
FLT3-ITD (positive vs. negative)	1.232	0.771	1.968	0.3839				
NPM1-mutated (positive vs. negative)	0.94	0.61	1.46	0.7907				
Karyotype risk group (Intermediate vs. Favorable)	2.05	1.14	3.68	0.0166	1.973	1.097	3.549	0.023
Karyotype risk group (Adverse vs. Favorable)	2.22	0.89	5.57	0.0891	2.301	0.917	5.776	0.076
Post-remission therapy (AuSCT vs. alloHCT)	1.4	0.84	2.32	0.1926				
Post-remission therapy (no SCT vs. alloHCT)	2.16	1.28	3.62	0.0037				

**Table 3 cancers-13-01083-t003:** Univariate and multivariate analysis for Disease Free Survival.

Parameter	Univariate Analysis				Multivariate Analysis			
	HR	Lower 95% CI	Higher 95% CI	*p*	HR	Lower 95% CI	Higher 95% CI	*p*
Treatment strategy (MRD-oriented vs. donor-oriented)	0.52	0.36	0.74	0.0004				
Sex (male vs. female)	1.15	0.79	1.66	0.4654				
Age (>60 vs. <60 years)	1.02	1	1.03	0.0615				
WBC count (× 10^9^/L) (>50.0 vs. <50.0)	1.596	1.0918	2.3339	0.0158				
FLT3-ITD (positive vs. negative)	1.253	0.812	1.933	0.3081				
NPM1-mutated (positive vs. negative)	0.88	0.58	1.33	0.5552				
Karyotype risk group (Intermediate vs. Favorable)	1.96	1.15	3.35	0.013	2.543	1.483	4.36	0.001
Karyotype risk group (Adverse vs. Favorable)	2.31	0.99	5.41	0.0535	3.853	1.596	9.302	0.003
Post-remission therapy (AuSCT vs. alloHCT)	1.67	1.03	2.73	0.0389	2.168	1.28	3.671	0.004
Post-remission therapy (no SCT vs. alloHCT)	3.2	1.95	5.24	<0.001	4.191	2.482	7.077	<0.001

## Data Availability

The data presented in this study are available in this article (and Appendix A).

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
