# Peer review of "Use of Measurable Residual Disease to Evolve Transplant Policy in Acute Myeloid Leukemia: A 20-Year Monocentric Observation"

_cancers, 2021, doi:10.3390/cancers13051083_

Round 1

Reviewer 1 Report

This article reports that authors demonstrated that the evolution to a comprehensive transplant selection policy, based on the integration of upfront genetics/cytogenetics and post-consolidation multiparametric flow cytometry MRD assessment (MRD-driven therapy) improves outcome of adult patients with AML as compared to a Donor-driven approach in their 20-year retrospective analysis. Moreover, it shows that MRD-driven therapy approach determines a superior outcome of intermediate-risk patients whereat in low-risk ones a careful evaluation is needed for transplant allocation.

Although the manuscript is well written, several points should be amended or clarified for the benefit of the readers.

Major considerations;

The usefulness of FCM-MRD based treatment strategy in AML has already reported from several clinical study groups. To deeply understanding, the authors should present the following results;

  1. The clinical characteristics of MRD-driven cohort patients (115) according to FCM-MRD level (Positive vs Negative).
  2. The results of OS, EFS, CIR, and NRM according to FCM-MRD level in each risk groups (LR, IR, HR).

Minor considerations:

  1. In Line 360; Figure 2A might be not correct. Figure 3C?
  2. In Line 363-364; Figure 2B might be not correct. Figure 3D?

Author Response

Dear editor

Please find enclosed the revised version of the manuscript entitled “Use of Measurable Residual Disease to evolve transplant policy in Acute Myeloid Leukemia: a 20-year monocentric experience” by Francesco Buccisano et al.

Our responses to the reviewers are structured on a point-by-point basis. Text modification and additions have been highlighted.

The usefulness of FCM-MRD based treatment strategy in AML has already reported from several clinical study groups. To deeply understanding, the authors should present the following results:

  • The clinical characteristics of MRD-driven cohort patients (115) according to FCM-MRD level (Positive vs Negative).

The analysis has been performed and the results are now mentioned in the text (224-230) and in a supplemental table (Table 1S)

  • The results of OS, EFS, CIR, and NRM according to FCM-MRD level in each risk groups (LR, IR, HR).

The analysis was performed and illustrated in the text (lines 287-290). Furthermore, a supplemental summary table was included (Table 2S).

Minor considerations:

  • In Line 360; Figure 2A might be not correct. Figure 3C?

Typo error. The text has been modified accordingly

  • In Line 363-364; Figure 2B might be not correct. Figure 3D?

Typo error. The text has been corrected accordingly

We hope that our manuscript in the revised version may meet your interest and that of Cancers readership.

Best regards

Francesco Buccisano

Reviewer 2 Report

The authors performed a retro-perspective analysis which investigated differences between donor-driven and MRD-driven transplant strategies for post-therapy AML. However, being retrospective, the study is merely indicative as guidance for future transplantation strategies. MRD as a  pre-transplant prognostic predictor has been shown previously.

On page 8 rows 282-283 the authors state that: “All the variables with a statistical significance in univariable analysis both for OS and DFS were challenged into a multivariable model (Table 2 and 3)”. However, WBC count and Post-remission therapy (no SCT vs. alloHCT) is missing from the analysis in Table 2 and Treatment strategy (MRD-oriented vs. donor-oriented), WBC count is missing from the analysis in Table 3. I suggest the authors to reanalyze their data with multivariate models without omitting these variables.

DFS is misspelled on page 6 row 239.

Author Response

Dear editor

Please find enclosed the revised version of the manuscript entitled “Use of Measurable Residual Disease to evolve transplant policy in Acute Myeloid Leukemia: a 20-year monocentric experience” by Francesco Buccisano et al.

Our responses to the reviewers are structured on a point-by-point basis. Text modification and additions have been highlighted.

  • On page 8 rows 282-283 the authors state that: “All the variables with a statistical significance in univariable analysis both for OS and DFS were challenged into a multivariable model (Table 2 and 3)”. However, WBC count and Post-remission therapy (no SCT vs. alloHCT) is missing from the analysis in Table 2 and Treatment strategy (MRD-oriented vs. donor-oriented), WBC count is missing from the analysis in Table 3. I suggest the authors to reanalyze their data with multivariate models without omitting these variables.

We thank the reviewer for the opportunity to clarify a statistical aspect that was not properly addressed in the original text. Variables included in the multivariate analysis were not only selected among factors with univariate association within p-value <0.1 but also by a step-by-step iterative construction that involves the selection of independent variables to be considered in the final model. This was done to avoid including variables whose collinearity may affect the results of the multivariable approach. We are confident that, with this further piece of information, the structure of the multivariate model might be better clarified but left unchanged. A sentence detailing these clarifications was added in the statistical (lines 194-199) and in the result (lines 292-293) sections.

  • DFS is misspelled on page 6 row 239.

The text has been corrected accordingly

We hope that our manuscript in the revised version may meet your interest and that of Cancers readership.

Best regards

Francesco Buccisano

Round 2

Reviewer 1 Report

Dear Authors,

Thank you for your efforts. In fact, some interesting results were obtained in the additional analysis. Based on these results, I could understand the strength of MRD-driven therapy in adult AML deeply.

Reviewer 2 Report

I am satisfied with the revisions made by the authors.